# Weight-Control Behaviors and Dietary Intake in Chinese Adults: An Analysis of Three National Surveys (2002–2015)

**DOI:** 10.3390/nu15061395

**Published:** 2023-03-14

**Authors:** Miyang Luo, Yixu Liu, Ping Ye, Shuya Cai, Zhenzhen Yao, Liyun Zhao, Jiayou Luo, Dongmei Yu

**Affiliations:** 1Xiangya School of Public Health, Central South University, Changsha 410008, China; miyangluo@csu.edu.cn (M.L.); yixu.liu@csu.edu.cn (Y.L.); y13317437129@163.com (Z.Y.); 2Yuxi Center for Disease Control and Prevention, Yuxi 653100, China; yep20227890@163.com; 3Key Laboratory of Trace Element Nutrition of National Health Commission, National Institute for Nutrition and Health, Chinese Center for Disease Control and Prevention, Beijing 100050, China; caisy@ninh.chinacdc.cn (S.C.); zhaoly@ninh.chinacdc.cn (L.Z.)

**Keywords:** weight control, dietary intake, diet quality, China healthy diet index, Chinese resident

## Abstract

Weight control through dietary management is becoming increasingly common worldwide. This study aimed to evaluate and compare the dietary intake and diet quality between Chinese adults with and without weight-control behaviors. Data were collected from the China National Nutrition Survey 2002, 2012, and 2015. Dietary intake was assessed using a combination of 24 h dietary recall of three consecutive days and a weighing method. Diet quality was calculated based on China healthy diet index (CHDI). A total of 167,355 subjects were included, of which 11,906 (8.0%) adults reported attempting to control weight within the past 12 months. Participants with weight control had lower daily total energy intake, as well as lower percentages of energy from carbohydrates, low-quality carbohydrates, and plant protein, but higher percentages of energy from protein, fat, high-quality carbohydrates, animal protein, saturated fatty acids, and monounsaturated fatty acids than those without weight control. Additionally, the CHDI score in the weight-control group was higher than those without (53.40 vs. 48.79, *p* < 0.001). Fewer than 40% of participants in both groups met the requirement for all specific food groups. Chinese adults who reported weight-control behaviors had an energy-restricted diet characterized by reduced carbohydrate intake and overall higher diet quality compared with those without dietary-control behaviors. However, both groups had significant room for improvement in meeting dietary recommendations.

## 1. Introduction

Overweight and obesity have become a public health concern in recent years, and weight control is an essential component for addressing this issue [1]. Effective weight control relies heavily on dietary management, which can be achieved through various energy-restricted and healthy dietary interventions such as low-fat, low-carbohydrate, and Mediterranean diets [2,3]. Despite the availability of such dietary options, individuals attempting weight control may lack the scientific knowledge to adopt favorable dietary behaviors [4]. Furthermore, even those who possess a clear understanding of healthy dietary principles may find it challenging to implement them in practice due to personal food preferences and an irresistible appetite [5,6].

Several previous studies have examined dietary intake among individuals attempting weight control [7,8,9,10]. These studies have shown that, regardless of the dietary management strategy used, people with weight-control attempts generally have lower energy intake compared with those without [7,8,9]. However, mixed findings have been reported regarding diet quality and adherence to dietary guidelines [11,12,13,14]. While some studies have observed improvements in diet quality among participants with weight control compared with those without [11,12], other studies have found no differences in refined grain, sodium, and cholesterol intakes or diet quality between populations with and without weight control [13,14]. Furthermore, some studies have even suggested that individuals with weight-control behaviors may be at increased risk of dietary inadequacy due to unhealthy dietary practices [7]. It is worth noting that most of these studies were conducted in developed countries or on people with overweight or obesity [7,8,15], and there is a particular lack of representative studies on the general population in China. Therefore, assessing dietary intake among populations with weight-control behaviors in China may help to identify the deficiencies in their dietary management practices and provide targeted guidance on dietary intake and weight management.

In this study, we used three rounds of nationally representative data from 31 provinces in China to learn the current status of dietary nutrition intake among Chinese adults with weight-control behaviors. We compared the dietary intake and diet quality between adults with and without weight-control behaviors and evaluated their adherence to dietary guidelines. This study provided a theoretical basis for improving the nutritional intake of people attempting weight control, promoting balanced diets, and encouraging the use of healthier and more scientific dietary management practices for weight control.

## 2. Materials and Methods

### 2.1. Study Population

This study was based on three consecutive rounds of the China National Nutrition Surveys (CNNS) in 2002, 2012, and 2015. CNNS is a national survey carried out by the Chinese Center for Disease Control and Prevention every 5–10 years since 1959, and multistage stratified cluster sampling methods were used to select residents across all provinces in China. The details of the survey design and methods have been described elsewhere [16]. Initially, 69,583, 67,177, and 96,631 subjects participated in the household dietary survey from 2002 to 2015. Participants in this study had the following inclusion criteria: (a) completed at least two days of the three-day dietary recall; (b) age greater than 18 years old; and (c) with complete personal information. Participants were excluded with the following criteria: (a) they were pregnant or breastfeeding; (b) with missing information on weight control behaviors; and (c) with extreme daily total energy intake, i.e., <800 kcal or >5000 kcal. The flow of inclusion is shown in Figure 1.

The series of nationwide surveys was approved by the Ethics Committee of the National Institute of Nutrition and Health, Chinese Center for Disease Control and Prevention (Approval number: 201519-B and approval date: 06/2015). All participants signed informed consent prior to the investigation.

### 2.2. Data Collection and Dietary Assessment

A questionnaire was conducted by trained researchers using face-to-face interviews in which sociodemographic characteristics and personal health status were surveyed. Height and weight were measured by trained researchers afterward. Sociodemographic characteristics included gender, age, educational level, occupation, marital status, and family income level; personal health status included weight-control behavior and a history of chronic diseases. Weight control was assessed using the question in the questionnaire “Have you tried to lose weight during the past one year?” If the answer was “yes”, the participant was considered to have weight-control behavior and was classified in the “yes” group of weight control; if the answer was “no”, the participant was considered to have no weight-control behavior and was classified in the “no” group of weight control.

Dietary intake was assessed using a combination of 24 h dietary recall of three consecutive days and a weighing method. Food intake during the past 24 h was recorded for each dietary recall day, including two weekdays and one weekend. The recall was assisted by the interviewer to ensure that accurate information was collected. Cooking oil and condiments were weighed at the household by researchers at the beginning and end of each 24 h dietary survey. Nutrient intakes were calculated using the China Food Composition tables (FCTs) [17,18]. The percentage of energy from carbohydrate, fat, and protein was calculated using the following equations: the total amount of carbohydrate intake × 4/the total energy intake × 100, the total amount of fat intake × 9/the total energy intake × 100, and the total amount of protein intake × 4/the total energy intake × 100, respectively. The recommended intake data of the percentage of energy intake from macronutrients were obtained from the Chinese dietary reference intakes—Part 1: Macronutrient (WS/T 578.1–2017) [19]. Specifically, 50–65% of total energy intake was recommended to come from carbohydrates, 10–15% from proteins, and 20–30% from fats [19]. We further subdivided proteins into plant and animal proteins; carbohydrates into low-quality and high-quality carbohydrates; and fats into monounsaturated, saturated, and polyunsaturated fatty acids, as detailed in a previous study [20]. The recommended daily food intake was derived from the 2016 Dietary Guidelines for Chinese Residents, and the recommendations were as follows: cereals and tubers 250–400 g/day; soybeans and nuts 25–35 g/day; vegetables 300–500 g/day; fruits 200–350 g/day; livestock and poultry meats 40–75 g/day; dairy products greater than 300 g/day; eggs 40–50 g/day; aquatic products 40–75 g/day; oil 25–30 g/day; and salt less than 6 g/day [21].

Diet quality was assessed using the China healthy diet index (CHDI) [22]. The CHDI includes 13 items with a total score of 100, and a higher score indicates better diet quality. The details of the scoring method are shown in Appendix A. Diet quality was categorized into two groups, i.e., satisfied and not satisfied, based on the CHDI score at the cutoff of 60. 

### 2.3. Statistical Analysis

The poststratification population sampling weights were derived from the sampling probability based on census data of 2010 in each survey round, and standardized data with adjustments for age and gender distribution were presented. Sociodemographic characteristics were compared between participants with and without weight control using chi-square tests. Intake of food and macronutrients and dietary score were described using means and 95% confidence intervals (CIs) with adjustment for the sample weights. General linear regression models were used to compare the difference in dietary intake and dietary score between groups, with adjustments for age, gender, education level, occupation, and family income level. Multivariate logistic regression was used to analyze the influence of weight-control behavior on diet quality. A two-sided *p* < 0.05 was considered to indicate statistical significance. Statistical analyses were conducted using SPSS 25.0 statistical software (IBM SPSS Inc., Chicago, IL, USA).

## 3. Results

### 3.1. Participant Characteristics

A total of 167,355 adults aged 18 to 107 were included in this analysis, with 45,148 in 2002, 53,578 in 2012, and 68,629 in 2015. Demographic characteristics were similar across participants from the three survey rounds (Appendix A). Overall, the prevalence of overweight plus obesity was 41.7%, and 8.0% of the included participants reported being on weight-control diets. The comparison of participant characteristics between adults with and without weight control behavior is shown in Table 1. Participants with weight-control behavior were more likely to be females, aged below 40 years old, had higher education levels and higher income levels, and lived in urban areas. The prevalence of overweight plus obesity in those with and without weight-control behavior was 70.4% and 39.2%, respectively.

### 3.2. Comparison of Dietary Intake in Participants with and without Weight-Control Behavior

As shown in Table 2, participants with weight-control behavior had lower average daily total energy intake compared with those without (1958.4 kcal/day vs. 2115.8 kcal/day, *p* < 0.001). They also showed a significantly lower percentage of energy intake from carbohydrates (51.9% vs. 55.6%, *p* < 0.001) and a higher percentage from protein (13.0% vs. 12.2%, *p* < 0.001) and fat (35.1% vs. 32.1%, *p* < 0.001), although the amount of total protein intake was slightly higher (63.3 g/day vs. 64.4 g/day, *p* < 0.001) compared with those without weight-control behavior. More specifically, participants with weight-control behavior showed a higher percentage of energy intake from high-quality carbohydrates (5.7% vs. 4.6%, *p* < 0.001) and a lower one from low-quality carbohydrates (47.5% vs. 52.0%, *p* < 0.001) than those without weight-control behavior. The percentage of energy intake from plant protein was higher than animal protein in both groups, and the weight-control group had a higher percentage of energy intake from animal protein (5.4% vs. 4.4%, *p* < 0.001) and lower energy intake from plant protein (7.0% vs. 7.5%, *p* < 0.001) than the group without weight-control behavior. In addition, the percentage of intake of both saturated fatty acids (9.1% vs. 8.6%, *p* = 0.009) and monounsaturated fatty acids (14.8% vs. 9.4%, *p* = 0.001) was higher than those without weight-control behavior.

In terms of food intake, lower intake of cereals and tubers (359.1 vs. 430.6 g/day, *p* < 0.001), vegetables (269.2 vs. 272.2 g/day, *p* = 0.003), oil (40.4 vs. 41.7 g/day, *p* < 0.001), and salt (12.1 vs. 13.2 g/day, *p* < 0.001) and higher intake of fruits (78.8 vs. 44.3 g/day, *p* < 0.001), meat (101.6 vs. 91.3 g/day, *p* = 0.045), dairy products (41.7 vs. 23.4 g/day, *p* < 0.001), eggs (30.8 vs. 23.9 g/day, *p* = 0.009), and aquatic products (39.7 vs. 30.9 g/day, *p* < 0.001) were observed in participants with weight-control behavior compared with those without.

### 3.3. Adherence to Dietary Recommendations in Participants with and without Weight-Control Behavior

In terms of macronutrients, around one-third of participants met the recommendation of carbohydrates and fats, and around 60% of participants fulfilled the requirement for protein (Figure 2). A higher percentage of participants met the requirement for carbohydrates in the weight-control group than the group without weight-control behavior (32.6% vs. 29.5%, *p* < 0.001), while the percentages were lower for protein (60.4% vs. 62.4%, *p* < 0.001) and fats (26.4% vs. 28.9%, *p* < 0.001). Moreover, a higher percentage of participants showed lower than the recommended level of carbohydrate intake in the weight-control group than in the no-weight-control group. Fewer than 40% of participants met the requirement for all specific food groups in both groups. More specifically, more than half of the participants consumed lower than recommended levels of soybeans and nuts, fruits, vegetables, dairy products, eggs, and aquatic products, while around 90% of participants consumed more salt than dietary guidelines prescribe.

### 3.4. Comparison of Diet Quality in Participants with and without Weight-Control Behavior

Overall, we observed that only 20.0% of participants had diet quality scores above 60 (Appendix A). The total diet quality score was 53.40 in the weight-control group and 48.79 in the other group (*p* < 0.001) (Table 3). Participant with weight-control behavior showed higher scores in food variety (6.73 vs. 5.01, *p* < 0.001), total vegetables (3.43 vs. 3.29, *p* < 0.001), dark vegetables (2.16 vs. 1.94, *p* < 0.001), fruits (3.04 vs. 1.69, *p* < 0.001), dairy products (1.63 vs. 0.85, *p* < 0.001), soybeans (3.72 vs. 3.20, *p* < 0.001), meat and eggs (4.08 vs. 3.56, *p* < 0.001), and aquatic products (1.88 vs. 1.43, *p* < 0.001) compared with those without weight-control behavior. The scores were lower among those with weight-control behavior in refined grains (4.84 vs. 4.93, *p* < 0.001), estimated percentage of energy from saturated fatty acid (8.52 vs. 8.58, *p* < 0.001), and sodium (4.98 vs. 5.03, *p* = 0.020) compared with those without weight-control behavior. Multivariate analysis showed participants with weight-control behavior had better diet quality scores after adjustment for demographic factors and BMI (Appendix A).

## 4. Discussion

In this study, we evaluated and compared the dietary intake and diet quality of participants with and without weight-control behavior using three consecutive rounds of national surveys in China. We found that 8% of subjects tried to control their weight within the last year. Those with weight-control behavior reported lower total energy intake, and the percentage of energy intake from carbohydrates was lower, while the percentage of energy intake from protein or fat was higher compared with those without weight-control behavior. Participants with weight-control behavior showed better diet quality; however, the overall adherence to dietary guidelines was below satisfactory in both groups.

The prevalence of self-reported weight-control attempts varies across different studies considering the difference in dietary patterns, sociodemographic characteristics, socioeconomic status, and the prevalence of overweight and obesity across countries [23]. A 2017 systematic review and meta-analysis of 72 studies showed that 34.6% of participants in the general population attempted to lose weight, while 24.7% attempted to maintain their weight [24]. However, there is limited information on the prevalence of weight-control attempts in China, particularly among the general population. Most existing studies are based on individuals with overweight or obesity [25,26]. For instance, a national survey on chronic disease surveillance found that 16.3% of adults with overweight or obesity attempted weight-control behaviors in China [25]. In our study, the prevalence of weight-control attempts in the general population was found to be 8.0%, while the prevalence was 12.0% among participants with overweight and obesity. The relatively low prevalence of weight-control attempts, especially among those with overweight and obesity in China, may be due to misconceptions about body weight and a lack of scientific knowledge on weight management [27,28,29]. It is therefore necessary to promote education on weight-control among the general population and provide targeted interventions for weight control among individuals with overweight or obesity to improve overall health. 

Dietary management for weight control involves various energy-restricted dietary interventions. In this study, we found that participants with weight-control behavior had a lower total energy intake compared with those without weight-control behavior, mainly by reducing carbohydrate intake. Previous studies also compared dietary intake between participants with and without weight-control behavior [30]. However, in some studies, the lower total energy intake among participants with weight-control behavior was accompanied by reduced fat intake, with no significant differences in carbohydrate intake [7,14,30,31,32]. This may reflect differences in dietary patterns for weight control among countries, as the intake of carbohydrates in China is relatively higher than many Western countries [33]. Our study also found that the percentages of energy from fat were higher in individuals with weight-control behavior than in those without, possibly because the weight-control individuals conducted an energy-restricted diet focused on reducing carbohydrate intake, resulting in a relatively higher proportion of energy intake from fat [34]. However, the proportion of energy intake from fat exceeded the recommended range in both groups, consistent with the results of previous studies [35]. This suggests that there is an unreasonable percentage of energy sources in the Chinese population, which is an important risk factor for obesity and other chronic diseases [36]. Therefore, it is essential to strengthen guidance on nutrition-related knowledge to help people scientifically adjust the composition of energy from macronutrients and promote their dietary structure in a more health-friendly direction. Regarding specific dietary intake, we found that participants with weight-control behavior showed a higher percentage of energy from high-quality carbohydrates and monounsaturated fatty acids and a lower percentage of energy from low-quality carbohydrates than those without weight-control behavior, indicating healthier eating habits. Moreover, participants with weight-control behavior had a higher percentage of energy from animal protein and slightly higher percentages of energy from saturated fatty acids, probably due to a higher intake of meats, eggs, and milk than those without. Plant protein has been associated with various benefits, including weight management [37]. Therefore, it is crucial to promote increasing the proportion of plant protein intake, such as increasing soybean and nut intake for Chinese adults with weight-control behavior, and increasing the proportion of high-quality carbohydrate intake for all Chinese adults.

Consistent with previous studies, participants with weight-control behavior in our study had a better diet quality than those without [11,12,13]. We observed improvement in food variety and found that participants with weight-control behavior had a higher score for intakes of vegetables, fruits, dairy products, soybeans, meat and eggs, and aquatic products compared with those without weight-control behavior. These differences between the two groups suggested that participants with weight-control behavior had a healthier and more balanced diet. However, there was no significant difference between the two groups in their whole grain intake, and weight controllers in China may consider increasing their intake of whole grains to achieve better weight loss, as whole grain intake can help improve weight management [38]. It is also important to note that the overall diet quality score was below an optimal level, and adherence to dietary guidelines was poor in both groups, suggesting that there is a need for improvement in both groups. Specifically, the average intake of fruits and dairy products was much lower than the recommended level, and more than two-thirds of participants consumed lower than recommended levels of fruits, soybeans and nuts, eggs, dairy products, and aquatic products, while more than 80% of participants consumed a higher level of salt than the dietary guidelines prescribe. These gaps in meeting the recommended levels of dietary guidelines among Chinese adults were also observed in previous studies [39,40]. These findings suggest that effective measures are needed to promote diet quality and adherence to dietary guidelines in the population. Additionally, our study indicated that gender, age, education level, occupational status, household income level, and urban/rural areas had varying degrees of influence on diet quality among Chinese adults, which is consistent with a prior study [41]. Therefore, it is necessary to improve the nutritional health literacy of residents in a targeted manner and increase the promotion of healthy dietary guidelines to achieve a healthier diet.

This study has several strengths, including the use of national representative data from 31 provinces in China, which allows for a more comprehensive evaluation of dietary intake and quality. We acknowledge that this study has the following limitations. Firstly, weight control can be achieved through various measures, including physical activity and medical treatment, and this study primarily focused on dietary interventions. Therefore, the results may not be applicable to individuals who utilize other weight-control methods. Additionally, self-reported weight-control behaviors may be subject to recall bias, and future studies should consider incorporating objective measures to verify weight-control status. Finally, dietary recall is also prone to recall bias, and future studies should consider incorporating additional measures, such as food diaries or biomarkers, to more accurately assess dietary intake.

## 5. Conclusions

To summarize, our study found a low prevalence of weight-control behavior among Chinese adults, particularly among those who were overweight or obese. We also observed that those with weight-control behavior implemented an energy-restricted diet focused on reducing carbohydrate intake and had better overall diet quality compared with those without weight-control behavior. However, there is still a significant need for improvement in the diet quality of both groups. Our study highlights the need for more effective strategies and measures to promote and refine the implementation of weight-control behavior in the Chinese adult population.

## Figures and Tables

**Figure 1 nutrients-15-01395-f001:**
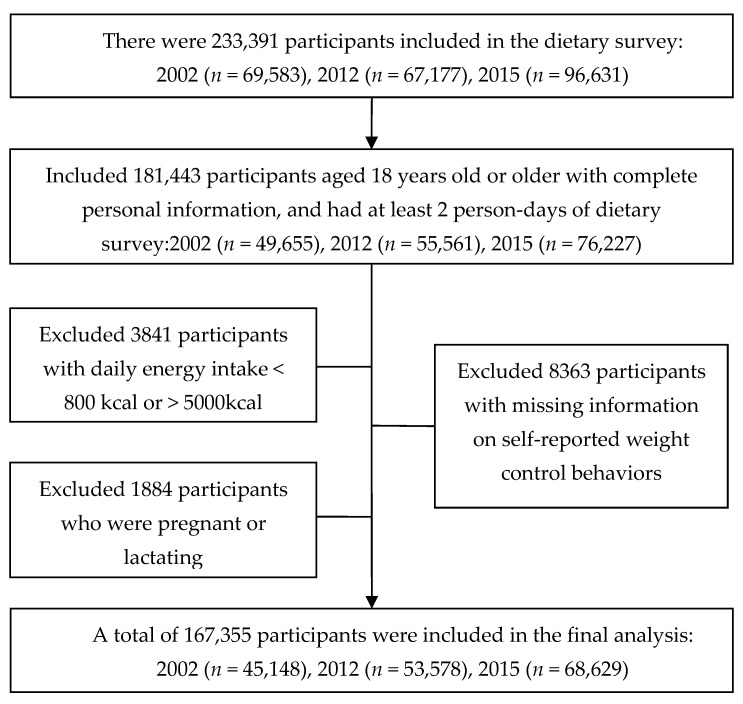
Flow of inclusion into the study.

**Figure 2 nutrients-15-01395-f002:**
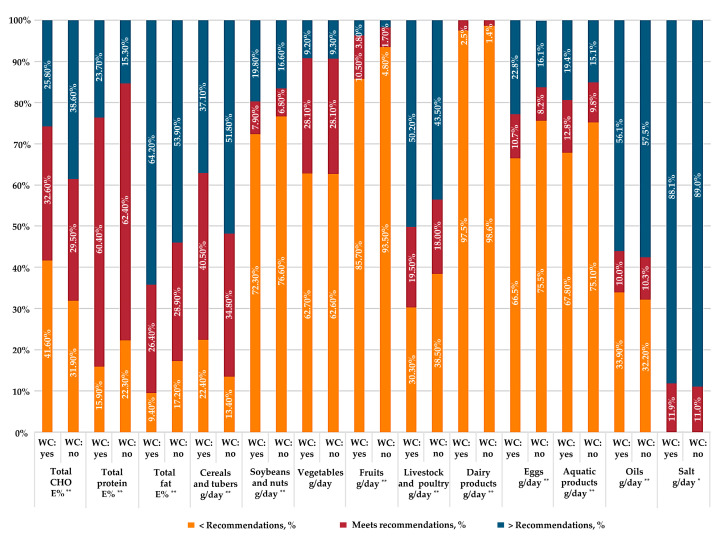
Percentage of Chinese adults complying with the dietary recommendations in the three rounds China National Nutrition Surveys. Dietary recommendations in the Dietary Guidelines for Chinese Residents (2016): 25–35 g/day soybeans and nuts; 250–400 g/day cereals and tubers; 300–500 g/day vegetables; 40–75 g/day livestock and poultry meats; 200–350 g/day fruits; 40–50 g/day eggs; 40–75 g/day aquatic products; >300 g/day dairy products; 25–30 g/day oil; <6 g/day salt. Dietary recommendations in the Chinese dietary reference intakes: 50–65 E% CHO, 10–15 E% protein, and 20–30 E% fat. Comparison of complying with the dietary recommendations between adults with and without weight-control behavior was conducted using the chi-square test. * *p* < 0.05, ** *p* < 0.001. Abbreviations: E%, percentage of energy; CHO, carbohydrate; WC: yes, yes weight control; WC: no, no weight control.

**Table 1 nutrients-15-01395-t001:** Characteristics of Chinese adults in the three rounds of China National Nutrition Surveys.

Variable	Overall *n* = 167,355	Weight Control: Yes *n* = 11,906	Weight Control: No *n* = 155,449	*p*
Gender				<0.001
Male	79,531 (50.6)	4731 (42.6)	74,800 (51.2)	
Female	87,824 (49.4)	7175 (57.4)	80,649 (48.8)	
Age group (years)				<0.001
18–39	43,488 (43.2)	4118 (54.7)	39,370 (42.2)	
40–64	94,051 (45.7)	6513 (40.0)	87,538 (46.2)	
≥65	29,816 (11.1)	1275 (5.3)	28,541 (11.6)	
Education level				<0.001
Elementary school and below	75,197 (37.2)	2449 (15.3)	72,748 (39.2)	
Secondary school	79,730 (52.6)	6934 (57.3)	72,796 (52.1)	
College and above	12,361 (10.2)	2521 (27.4)	9840 (8.7)	
Occupation				<0.001
Employed	114,622 (72.2)	7420 (68.9)	107,202 (72.5)	
Unemployed	52,655 (27.8)	4482 (31.1)	48,173 (27.5)	
Annual income per capita				<0.001
Low	84,758 (55.4)	3558 (33.4)	81,200 (57.3)	
High	67,883 (44.6)	7363 (66.6)	60,520 (42.7)	
Area				<0.001
Urban	68,954 (40.6)	8314 (67.9)	60,640 (38.2)	
Rural	98,401 (59.4)	3592 (32.1)	94,809 (61.8)	
Body mass index (kg/m²)				<0.001
<18.5	7373 (5.7)	98 (1.2)	7275 (6.1)	
18.5–23.9	78,492 (52.6)	2705 (28.4)	75,787 (54.7)	
24.0–27.9	50,047 (30.1)	4774 (40.4)	45,273 (29.2)	
≥28.0	19,024 (11.6)	3506 (30.0)	15,518 (10.0)	

Data are *n* (weighted%). Number missing: occupation (*n* = 78), annual income per capita (*n* = 14,714), education level (*n* = 67), body mass index (*n* = 12,419). Comparison of characteristics between Chinese adults with and without weight-control behavior was conducted using the chi-square test.

**Table 2 nutrients-15-01395-t002:** Mean intake of dietary-related items among Chinese adults in the three rounds of China National Nutrition Surveys.

Variable	Recommendations ^1^	Weighted Mean (95%CI)		*p* ^2^
		Weight Control: Yes	Weight Control: No	
Energy and nutrients ^3^				
Energy intake, kcal/day		1958.4	2115.8	<0.001
		(1947.3–1969.6)	(2112.2–2119.4)	
Carbohydrate intake, g/day		253.9	296.3	<0.001
		(252.1–255.7)	(295.6–296.9)	
Protein intake, g/day		63.3	64.0	<0.001
		(62.8–63.7)	(63.8–64.1)	
Fat intake, g/day		76.6	75.0	0.003
		(75.9–77.3)	(74.8–75.2)	
Total carbohydrates, E%	50–65	51.9	55.6	<0.001
		(51.7–52.1)	(55.6–55.7)	
High-quality carbohydrates, E%		5.7	4.6	<0.001
		(5.6–5.8)	(4.5–4.6)	
Low-quality carbohydrates, E%		47.5	52.0	<0.001
		(47.3–47.7)	(51.9–52.1)	
Total protein, E%	10–15	13.0	12.2	<0.001
		(13.0–13.1)	(12.2–12.3)	
Animal protein, E%		5.4	4.4	<0.001
		(5.3–5.4)	(4.3–4.4)	
Plant protein, E%		7.0	7.5	<0.001
		(6.9–7.0)	(7.5–7.5)	
Total fat, E%	20–30	35.1	32.1	<0.001
		(34.9–35.3)	(32.1–32.2)	
Saturated fatty acids, E%		9.1	8.6	0.009
		(9.1–9.2)	(8.6–8.7)	
Monounsaturated fatty acids, E%		14.8	9.4	0.001
		(14.6–14.9)	(9.4–9.4)	
Polyunsaturated fatty acids, E%		10.8	13.9	0.064
		(10.7–10.9)	(13.9–14.0)	
Food groups				
Cereals and tubers, g/day	250–400	359.1	430.6	<0.001
		(356.1–361.8)	(429.3–431.4)	
Soybeans and nuts, g/day	25–35	19.7	17.3	<0.001
		(19.1–20.2)	(17.2–17.5)	
Vegetables, g/day	300–500	269.2	272.2	0.003
		(266.3–272.0)	(271.4–273.1)	
Fruits, g/day	200–350	78.8	44.3	<0.001
		(76.6–81.0)	(43.8–44.8)	
Livestock and poultry meats, g/day	40–75	101.6	91.3	0.045
		(100.0–103.2)	(90.8–91.8)	
Dairy products, g/day	>300	41.7	23.4	<0.001
		(39.7–43.8)	(22.9–23.9)	
Eggs, g/day	40–50	30.8	23.9	0.009
		(30.2–31.4)	(23.8–24.1)	
Aquatic products, g/day	40–75	39.7	30.9	<0.001
		(38.6–40.9)	(30.6–31.3)	
Oil, g/day	25–30	40.4	41.7	<0.001
		(39.8–40.9)	(41.5–41.8)	
Salt, g/day	<6	12.1	13.2	<0.001
		(12.0–12.3)	(13.1–13.2)	

The data were weighted to be nationally representative. ^1^ Levels of recommendations were based on the Dietary Guidelines for Chinese Residents (2016) and Chinese dietary reference intakes. ^2^ The *p* value was obtained using general linear models adjusted for gender, age, body mass index, area, education level, occupation, and annual income per capita. ^3^ High-quality carbohydrates were defined as carbohydrates from fruits, whole grains, nonstarchy vegetable, and legumes. Low-quality carbohydrates were defined as carbohydrates from added sugars, refined grains, other starchy vegetables, tubers, and other sources. Animal protein was defined as protein from livestock and poultry meats, aquatic products, eggs, dairy products, and other sources. Plant protein was defined as protein from refined grains, whole grains, nuts, legumes, and other sources.

**Table 3 nutrients-15-01395-t003:** CHDI components and criteria for scoring and the score of Chinese adults in the three rounds of China National Nutrition Surveys.

CHDI Component	Score Range	Standard for Maximum Score	Standard for Minimum Score of Zero	Weighted Mean (95%CI)		*p* ^1^
				Weight Control: Yes	Weight Control: No	
Food variety	0–10	≥12 kind	≤5 kind	6.73 (6.68–6.78)	5.01 (5.00–5.03)	<0.001
Refined grains	0–5	≥100 g/1000 kcal	0	4.84 (4.83–4.85)	4.93 (4.92–4.93)	<0.001
Whole grain, dry bean, and tuber	0–5	≥40 g/1000 kcal	0	2.08 (2.04–2.11)	2.09 (2.08–2.10)	0.392
Total vegetables	0–5	≥180 g/1000 kcal	0	3.43 (3.41–3.45)	3.29 (3.28–3.30)	<0.001
Dark green and orange vegetables	0–5	≥90 g/1000 kcal	0	2.16 (2.13–2.19)	1.94 (1.93–1.95)	<0.001
Fruit	0–10	≥110 g/1000 kcal	0	3.04 (2.98–3.11)	1.69 (1.67–1.70)	<0.001
Dairy	0–10	≥100 g/1000 kcal	0	1.63 (1.57–1.68)	0.85 (0.84–0.86)	<0.001
Soybean	0–10	≥10 g/1000 kcal	0	3.72 (3.65–3.79)	3.20 (3.18–3.22)	<0.001
Meat and egg	0–5	≥50 g/1000 kcal	0	4.08 (4.05–4.10)	3.56 (3.55–3.56)	<0.001
Fish, shellfish, and mollusk	0–5	≥30 g/1000 kcal	0	1.88 (1.85–1.92)	1.43 (1.42–1.44)	<0.001
Calories from SFAs	0–10	<10%	≥15%	8.52 (8.47–8.56)	8.58 (8.56–8.59)	<0.001
Sodium	0–10	≤1 g/1000 kcal	≥4 g/1000 kcal	4.98 (4.92–5.04)	5.03 (5.01–5.05)	0.020
Empty calories	0–10	≤20%	≥40%	7.94 (7.88–7.99)	8.05 (8.04–8.07)	0.407
Total	0–100			53.40 (53.20–53.60)	48.79 (48.73–48.85)	<0.001

The data were weighted to be nationally representative. Abbreviations: CHDI, China healthy diet index. SFAs, saturated fatty acids. ^1^ The *p* value was obtained using general linear models adjusted for gender, age, body mass index, area, education level, occupation, and annual income per capita.

## Data Availability

The data are not allowed to be disclosed according to the National Institute for Nutrition and Health and the Chinese Center for Disease Control and Prevention.

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
