# Peer review of "Weight-Control Behaviors and Dietary Intake in Chinese Adults: An Analysis of Three National Surveys (2002–2015)"

_nutrients, 2023, doi:10.3390/nu15061395_

Round 1

Reviewer 1 Report

Nutrients-2269012, by M Luo, is an interesting study of adult Chinese population admitted to a National Nutrition Survey, “aimed to evaluate and compare the dietary intake and diet quality” between those with and without weight control. The major limitation of this study is the incomparability of physical activity in the matched groups.

I suggest to Authors:

- Given that the study considers non-consecutive years (2002, 2012, 2015), even if there is a refer to a previous published study, add a comment about.

- At line 142-143 “The prevalence of overweight and obesity in those with and without weigh control was 70.4% and 39.2%, respectively."

Please let the lecture clearer, explain if "overweight and obesity" means "overweight plus obesity".

- In Table 1 "Weight control: yes n = 111.906" is an obvious mistake. Please accurately revise the whole table since at bottom the count of patients in “Body mass index” does not add up.

Author Response

请参阅附件。

Reviewer 2 Report

Methods -Characterized the weight control group (which individuals are considered to have weight control measures? or parametres?; clarify how was an individual sampled in this group)
Results - The tables should be mentioned in the text.
Discussion - should be organized by the subitems of the results and should be enriched with more references.
References - can be enriched and improved.

Round 2

Reviewer 1 Report

The current version is adequately corrected.